# Forty Years of Soil and Water Conservation Policy, Implementation, Research and Development in Indonesia: A Review

Hunggul Yudono Setio Hadi Nugroho [1,*], Tyas Mutiara Basuki [1], Irfan Budi Pramono [1], Endang Savitri [1], Purwanto [1], Dewi Retna Indrawati [1], Nining Wahyuningrum [1], Rahardyan Nugroho Adi [1], Yonky Indrajaya [1], Agung Budi Supangat [1], Pamungkas Buana Putra [1], Diah Auliyani [1], Eko Priyanto [1], Tri Wira Yuwati [2], Pratiwi [3], Budi Hadi Narendra [3], Asep Sukmana [3], Wuri Handayani [4], Ogi Setiawan [5] and Ryke Nandini [5]

1   Watershed Management Technology Center, Jl. Ahmad Yani, Surakarta 57102, Indonesia;
    tmbasuki@yahoo.com (T.M.B.); ibpramono@yahoo.com (I.B.P.); savitriendang@gmail.com (E.S.);
    purwanto_fris@yahoo.com (P.); dw_indrawati@yahoo.com (D.R.I.); nining0709@yahoo.com (N.W.);
    dd11lb@yahoo.com (R.N.A.); yonky_indrajaya@yahoo.com (Y.I.); maz_goenk@yahoo.com (A.B.S.);
    pamungkas_buanaputra@yahoo.co.id (P.B.P.); d_auliyani@yahoo.com (D.A.); eko.forest@gmail.com (E.P.)
2   Environment and Forestry Research and Development Institute of Banjarbaru, Jl. Ahmad Yani Km 28.7,
    Banjarbaru 70721, Indonesia; yuwatitriwira@gmail.com
3   Forest Research and Development Center, The Ministry of Environment and Forestry, Jl. Gunung Batu No. 5,
    Bogor 16118, Indonesia; pratiwi.lala@yahoo.com (P.); narendra17511@gmail.com (B.H.N.);
    asepsukmana@gmail.com (A.S.)
4   Agroforestry Research and Development Centre (ARDC), Ministry of Environment and Forestry, Jalan Raya
    Ciamis-Banjar Km 4, Ciamis 46201, Indonesia; ninikiank@gmail.com
5   Non-Timber Forest Products Technology Research and Development Institute, Jl. Dharma Bakti No. 7,
    Mataram 83371, Indonesia; o_setiawan@yahoo.com (O.S.); rykenand@yahoo.com (R.N.)
*   Correspondence: hunggulys@yahoo.com; Tel.: +62-8114093999

**Abstract:** Dominated by mountainous topography, high rainfall, and erosion-sensitive soil types, and with the majority of its population living in rural areas as farmers, most of Indonesia's watersheds are highly vulnerable to erosion. In 1984, the Government of Indonesia established 22 priority watersheds to be handled, which marked the start of formal soil and water conservation activities. Although it has not fully succeeded in improving watershed conditions from all aspects, something which is indicated by fluctuations in the area of degraded land, over the past 40 years the Indonesian government has systematically implemented various soil and water conservation techniques in various areas with the support of policies, laws and regulations, and research and development. These systematic efforts have shown positive results, with a 40% reduction in the area of degraded land over the last 15 years from 2004–2018. This paper reviews policy, implementation, and research and development of soil and water conservation activities in Indonesia over the last 40 years from the 1980s to 2020 and explores the dynamics of the activities.

**Keywords:** soil and water conservation; watershed; erosion

## 1. Introduction

Soil degradation has been a global problem [1,2] for millennia [2–4]. Degraded soil impacts not only the soil system but also the hydrological system. Soil erosion leads to the prominent deterioration of land quality, financial loss, and environmental services. Disastrous soil erosion and other soil degradation forms are major drawbacks for the sustainability of natural resources and the environment [4]. Global impacts of soil erosion decreased agri-food yield by 33.7 million tonnes, with a consequent increase in prices by 0.4 to 3.5% and increased water abstraction by 48 billion m$^3$ [5].

Indonesia experiences severe soil erosion due to its location in a tropically warm and humid climate with high population pressure. On average, annual rainfall is 2702 mm [6]

and reaches 5000 mm in mountain regions [7]. Of its total land area, Indonesia is dominated by sloping areas, with steeply dissected hilly and mountainous land covering 35 to 40%, level to undulating areas 30 to 35%, and the rest being coastal alluvial and peatlands [8]. Soil erosion ranges from 97.5 to 423.6 tonnes·ha$^{-1}$·year$^{-1}$ [9], causing output losses among some crops, including rice, cereals, horticultural, oilseeds, and sugar, of 4,102,000 tonnes·year$^{-1}$ [5]. In addition to geographical factors, Indonesia's large population also complicates the issue. The population of Indonesia was around 270 million people in 2020, and the number of people working in the agriculture, forestry, and fisheries sectors was 37,130,676, or 28.3% [10]. Sumiahadi and Acar [11] have stated that agricultural land is the major contributor to soil erosion in Indonesia. More than 50% of farmers have agricultural land ownership of less than 0.5 ha [12]. Therefore, people tend to use sloping areas for intensive cultivation to fulfil their needs, which induces severe soil erosion.

Accelerated soil erosion in Indonesia has occurred since the Dutch colonial era due to deforestation for forced cultivation of industrial crops known as the *Cultuur Stelsel* program [13]. Mass environmental improvement movements started several years after Indonesia's independence. However, integrated soil and water conservation began in the late 1970s and early 1980s through the Presidential Instruction on Afforestation and Reforestation, which was strengthened in its direction in the handling of soil conservation in 22 priority watersheds. The importance of the Soil and Water Conservation (SWC) received greater attention from the government after the massive flood in Solo, Central Java in 1966. SWC was implemented through a rehabilitation project in 1967 led by the Ministry of Agriculture.

Figure 1 illustrates the important momentum in Indonesia from the 1950s to 2020 related to soil and water conservation programs.

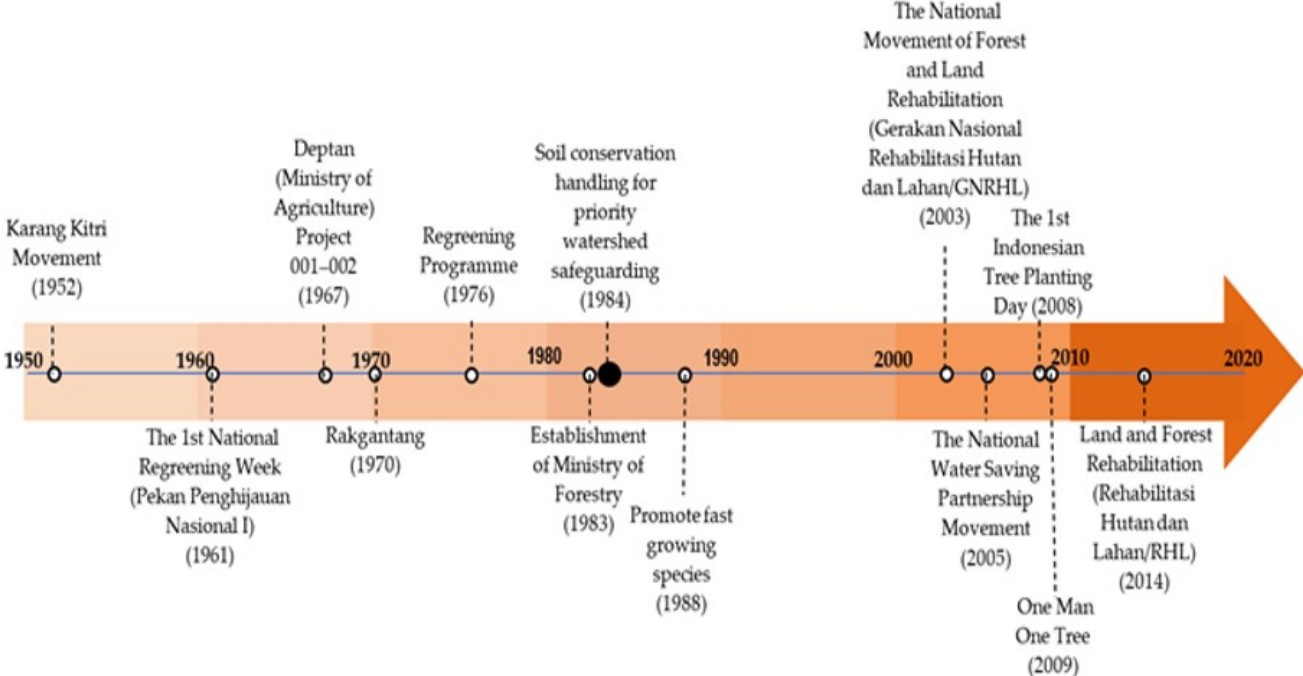

**Figure 1.** Milestones of soil and water conservation-related programs in Indonesia.

This paper is intended to review the implementation of SWC in Indonesia over four decades (1980 to 2020). The review covers research findings and experiences from the plot to the national scale in various ecoregions. The review source materials are nationally and internationally published research papers, research reports, relevant books, and field activities carried out by the authors.

## 2. Degraded Land Area in Indonesia for the Last Forty Years

In Law No. 37, 2014 on soil and water conservation, degraded land is defined as land that does not function properly as a production medium for growing cultivated or uncultivated plants. For decades, the government has carried out land rehabilitation and soil and water conservation throughout Indonesia to improve and curb the expansion of degraded land. Although soil and water conservation activities have been systematically carried out for decades, the area of degraded land in 2018 was still high at around 14 million hectares [14], even higher than the data for the previous 40 years, which show an area less than 10 million hectares between 1980 and 1994 [15]. Something that was due to the imbalance between the rate of land degradation and its recovery capacity. Figure 2 depicts the area of degraded land in Indonesia during the last forty years, collected from the Forestry Statistics Book 1980–2019.

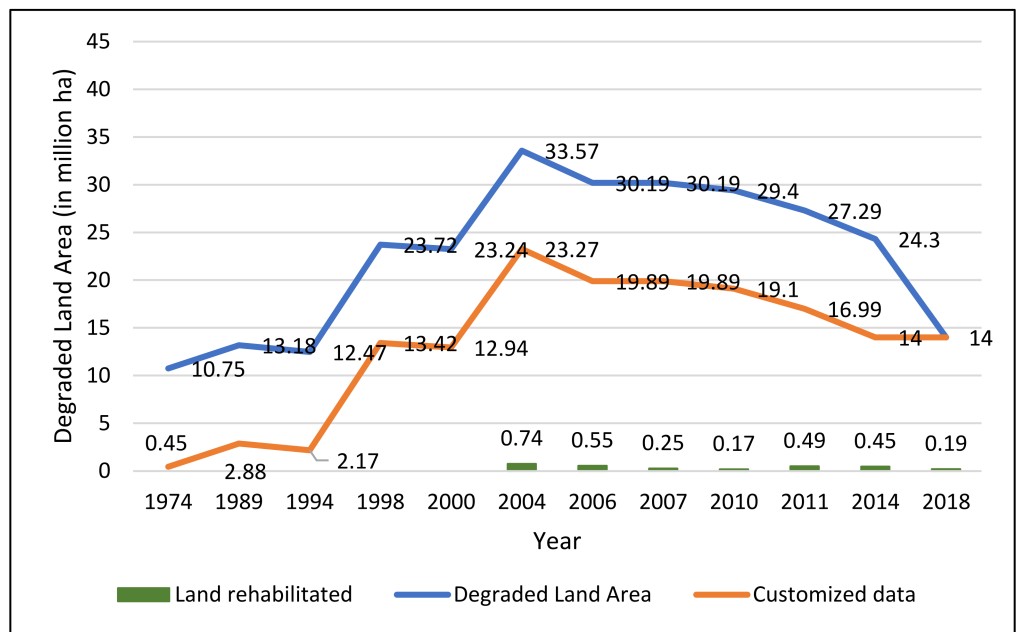

**Figure 2.** The Trend of degraded land in Indonesia from 1974 to 2018.

It appears that the area of degraded land increased from 1974 until 2004 and continued to decline until the last data in 2018. However, the drastic decline from 2014 to 2018, amounting to 10.3 million hectares, is not solely because of the success of Forest and Land Rehabilitation (RHL) activities, but because of changes in the criteria for determining degraded land used by the Ministry of Environment and Forestry [16]. In the new criteria, savanna and mountain craters are excluded from the status of degraded land. In addition, not all land with steep slopes is included in the degraded land criteria as the previous criteria. The amendment to the criteria for degraded land refers to Law No. 37, 2014, on soil and water conservation. In Figure 2, the blue line shows the trend of degraded land obtained from the statistics book, while the orange line represents the trend of degraded land data from the statistics (1974–2014) which have been reduced by 10.3 million hectares.

Figure 2 shows that the government's RHL activities from year to year appear to be very small proportionally compared to the area of degraded land. Therefore, when the area of degraded land consistently declined from 2004 to 2018 with a deviation much larger than the accumulated area of RHL, this then raises the question: What are the factors that reduce the rate of degraded land other than the government's official RHL activities? This will be discussed in the "closing notes" section at the end of this paper. The spatial distribution of the degraded land in Indonesia is presented in Figure 3.

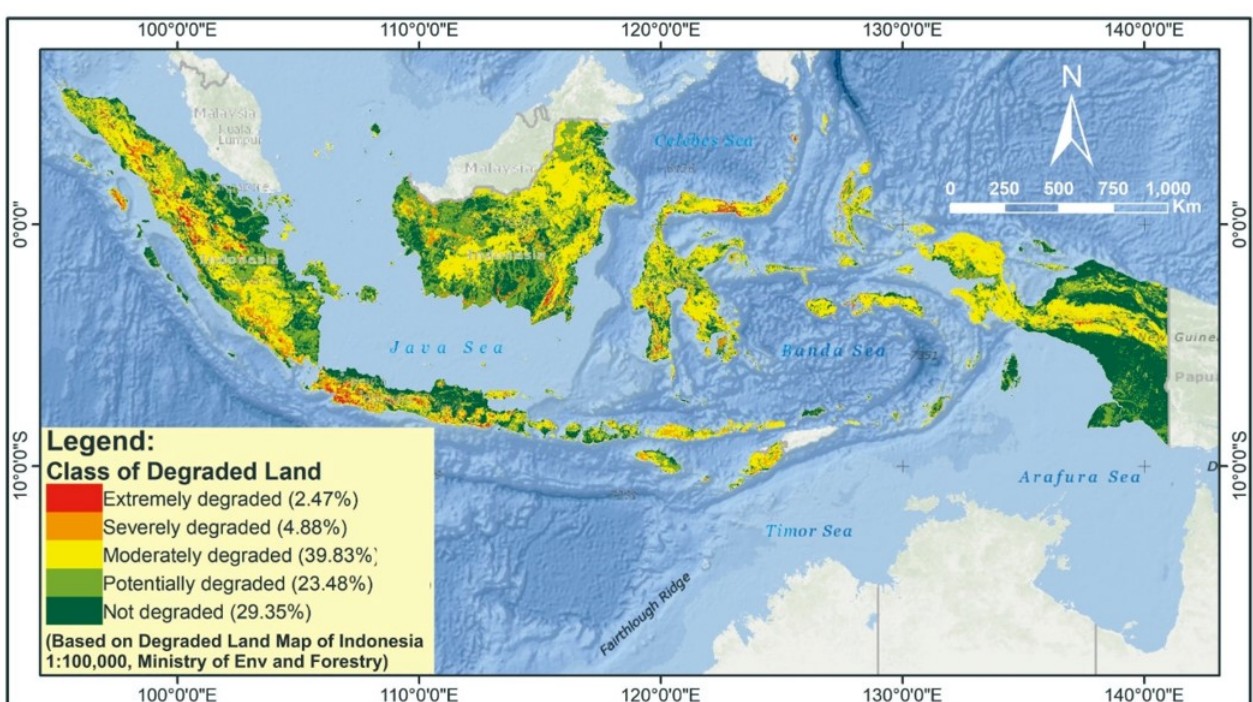

**Figure 3.** Spatial distribution of degraded land in Indonesia 2018.

## 3. Regulation, Institution, and Policy of SWC

### 3.1. Regulation and Rules

Based on the 1945 State Constitution, the state controls the land, water, and other natural resources to achieve people's prosperity and sustainability. Since 1960, several laws and regulations have stated the importance of the SWC. Law No. 5, 1960 mandates the importance of maintaining soil, including increasing its fertility and preventing its damage [17]. In the forestry sector, the importance of SWC through sustainable forest management has been mandated in Law No. 5, 1967 on Forestry. This regulation emphasizes the importance of water management to prevent floods and erosion while maintaining soil fertility [18]. In 1974, the government re-emphasized the importance of carrying out SWC based on law No. 11, 1974 on Irrigation. In 1976, a Presidential Decree (Inpres program) for Regreening and Reforestation was enforced as guidance for SWC activities for the whole country [8].

To emphasize the importance of SWC, a joint decree of the Minister of Forestry, Minister of Public Works, and Minister of Home Affairs No. 19, 1984, Kh. 059/Kpts-Ii/1984 and Pu.124/Kpts/1984 was issued on 4 April 1984. This decree focused on implementing SWC in the 22 Super Priority Watersheds. During the development of SWC implementation, in the 1980s and 1990s Usaha Pelestarian Sumber Daya Alam or Efforts for Natural Resources Conservation (UPSA), demonstration plots were built to provide direct guidance for SWC to communities. This UPSA program gradually disappeared and was reactivated through regulation No. SE. 6/PDASHL/SET/DA.1/9/2019 by the Directorate General of Watershed Management and Protected Forests or PDASHL. The highest level of SWC regulation is Law No. 37, 2014 on Soil and Water Conservation, which stipulates planning, implementation, and supervision of SWC, including rights and obligations as well as financing [19]. However, there has been no government regulation following the law until now. The State Government, by sector through the relevant ministries, regulates the implementation of SWC, emphasizing each sector. For example, the forestry sector emphasizes soil conservation activities as part of forest and land rehabilitation activities. Its activities are regulated in Government Regulation No. 26, 2020 on Forest Rehabilitation and Reclamation, which is a derivative of Law No. 41, 1999, and through Ministerial Regulation No. 105, 2018 on Forest and Land Rehabilitation. In the agricultural sector, there is Regula-

tion of Ministry of Agriculture No. 47/Permentan/OT.140/10/2006 on General Guidelines for Agricultural Cultivation in Mountain Land. In the mining sector, soil conservation activities are integrated with ex-mining reclamation activities as regulated in the Ministerial Regulation of Energy and Mineral Resources No. 7, 2014 [20], mainly for erosion control and soil fertility improvement [21]. The Ministry of Public Work and Housing, through Law No. 17, 2019 on Water Resources [22], emphasizes water conservation aspects.

The SWC implementation should be integrated among sectors and is part of watershed management. Technical regulations are needed starting from the planning, implementation, and financing stages. Institutionally, the mandate for water and soil conservation is attached to watershed management for the reason that the watershed is considered the most representative unit in the protection and maintenance of land functions. Therefore, the context of soil and water conservation basically increases the capacity and carrying capacity of the watershed. However, at the regulatory level, there is juridical confusion between Government Regulation No. 37, 2014 on Soil Conservation and Law No. 37, 2012 on Watershed Management. Soil conservation is essentially a part of watershed management activities. However, soil conservation is regulated by law in the regulatory hierarchy, while watershed management is only regulated at the government regulation level [20].

### 3.2. Soil and Water Conservation Institution

There are three categories of institutions of SWC in Indonesia, namely: 1, government institutions formed under the mandate of laws and other regulations (formal institution); 2, institutions formed by the government, consisting of NGO administrators, academics, researchers, and environmentalists who are concerned with SWC activities (semi-formal institutions); and 3, institutions formed by the community (informal institutions).

Ministerial-level government organizations directly related to SWC are the Ministry of National Development Planning, the Ministry of Forestry (MoF), the Ministry of Public Works and Housing, the Ministry of Home Affairs, and the Ministry of Agriculture. The main task of the Ministry of National Development Planning is the formulation of a national development plan that coordinates and integrates sectoral development plans, including forestry and water resource conservation. Based on Government Regulation No. 37, 2012 on Watershed Management, the MoF is mandated to conduct inter-provincial watershed, establish a watershed information center in each province, establish a watershed observer forum, and monitor and evaluate watershed performance [23]. Based on Law No. 17, 2019 on Water Resources, the Ministry of Public Works and Housing has the task of developing management and strategy for utilizing water resources, managing rivers and controlling the destructive power of river water, management of irrigation and dams, management of swamps and lakes, and provision of groundwater and raw water [24]. Based on Law No. 23, 2014 on Regional Government, provincial governments are mandated to manage watersheds and the environment within their administrative areas [25]. At the Ministry of Agriculture, the implementation of SWC activities is part of the Directorate General of Agricultural Infrastructure and Facilities authority. SWC technology is developed by the Indonesian Agency for Agricultural Research and Development, in particular the Indonesian Center for Agricultural Land Resource Research Development. SWC activities are mainly carried out on agricultural land to maintain the sustainability of farming resources through intensive and productive land protection, improving marginal land conditions, and increasing water availability [26].

Since the establishment of the MoF in 1983, SWC in the forestry sector has been conducted by the Directorate General of Reforestation and Land Rehabilitation to prepare a watershed management plan, while the Ministry of Public Works prepares a watershed management plan based on irrigation interests. These two plans should be approved by the Governor concerned. Once the planning documents are approved, they become a reference for other sectors. In fact, other sectors pay less attention to both plans due to some overlaps issues.

In relation to watershed management institutions (which also includes SWC activities), there are two related laws that are not in line. Law No. 23, 2014 on Regional Government states that watershed management is the responsibility of the central government and provincial governments. Meanwhile, Law No. 37, 2014 on Water and Soil Conservation (promulgated in the same year) and Government Regulation No. 37, 2012 on Watershed Management stipulate that watershed management is not only the authority of the central and provincial governments but also districts [20].

In addition to the above institutions, there are several supporting research and development institutions and organizations, namely, Watershed Management Technology Center (WMTC), Center for Research and Development of Water Resources (CRDWR), and Center for Soil Research (CSR). WMTC conducts research and development, which produces technology to support soil and water conservation activities, especially those carried out by the Ministry of Environment and Forestry. CRDWR is a supporting organization for water resource management carried out by the Ministry of Public Works, while CSR is a research institution under the Ministry of Agriculture.

There are watershed forums at the national, regional, and watershed unit levels in semi-formal institutions formed by the central government and local governments. In general, this forum has the function of reviewing policies, plans, and programs that are being and will be implemented in watershed management as well as problems that arise as a result of watershed management activities and natural disasters. This forum provides considerations and suggestions for solving problems for the government.

At the level of informal institutions, many non-governmental organizations act as drivers of community groups in the implementation of SWC techniques. Some examples of SWC institutions that emerge from below include (1) Farmer groups implementing SWC [27], (2) Water user community groups, and (3) community forest management groups in Wonogiri in Central Java, Gunungkidul in the Special Region of Yogyakarta and Krui-Liwa, in Lampung and Dolok Sanggul in North Sumatera etc. [28,29].

The participation of various parties, including the community, in watershed management, has been mandated in Government Regulation No. 37, 2012 on Watershed Management. However, at the field level, the synergy between formal, semi-formal, and informal institutions has not worked well.

### 3.3. Policy Dynamics

de Graaff, Aklilu, Ouessar, Asins-Velis and Kessler [6] have reported three common aspects in the policy development of SWC in some countries, including Indonesia. Firstly, different political regimes put in place different policies and methods for SWC, secondly, reforestation and terracing received much attention in the early period, and thirdly, forestry agencies or ministries were in control in most situations. A top-down strategy predominated in the first period, both from an institutional and geomorphological perspective. The SWC method has evolved through some phases to address the problem and improve actual livelihood and environmental effects [30,31]. These approaches can be divided into three phases: top-down interventions, populist or farmer-first, and neo-liberal approaches [30,31].

The first phase, which started in the 1950s and lasted roughly until the 1990s, focused on "top-down" terracing and tree planting [31,32]. This was also known as "the command-and-control strategy." In the period before 1965, during Indonesia's initial period of socialism, which some refer to as communism, the original leftist rule applied with some characteristics, e.g., land reforms were implemented, major holdings were nationalized, and different forms of farming activities and SWC measures were "enforced." In this period, the top-down approach for SWC was used mainly for promoting terraces and gully control. From 1966 to 1998, Indonesia was governed by a right-wing authoritarian regime, which likewise practiced top-down SWC. The focus was always on the (well terraced) rice industry in the lowlands, which was aided by financial, market, and institutional incentives in the 1970s and 1980s, whereas the uplands were not. The institutions involved were six ministries supported by some provincial and district governments.

Although still governed by the authoritarian regime, the emphasis of SWC development was shifted to a more "populist" approach in the late 1980s, influenced by Farmer First [33], emphasizing bottom-up techniques and dealing with fast rural evaluations, as well as village-level mapping and planning as a learning process [34]. In the process of technology development and extension, this strategy emphasized small-scale and bottom-up participatory interventions, typically using indigenous technologies [35], and essentially rejected the standard transfer of technology (TOT) model.

After 1998, the political regime shifted to a democratic regime where some authorities were transferred to local governments. In 2014, Indonesia established a law on SWC for the first time. Even though the law has not been followed by the lower regulations yet (i.e., government regulation), the law is an umbrella for SWC to be practiced better in the future. Due to the constant, slow, and almost imperceptible process of soil erosion, neither farmers nor policymakers place a high value on SWC issues in their everyday decision-making. However, inevitable catastrophes, particularly large floods, can significantly increase SWC efforts. For example, the government launched several significant donor-assisted watershed management initiatives (e.g., Solo and Citanduy) after the city of Solo (Surakarta) was severely inundated in 1966 [20,36].

Land rehabilitation has been a major emphasis since the commencement of the Five-Year Development Plan (Repelita) cycles in 1969, with the catchment area serving as the management unit. The Presidential Instruction (INPRES) on Reforestation and Regreening in 1969 was supported by various projects, including (1) The Citanduy I project (1976), in the downstream of the Citanduy watershed for flood management and irrigation, and Citandui II (1981–1988) project in the upper Citanduy watershed in 1981–1988 [37]; (2) the Upland Agriculture on Conservation Project (UACP) in the Jratun Seluna (Salatiga-Central Java) and Brantas watersheds (Malang-East Java) in 1984–1992; (3) the Upland Farmer Development Project (UFDP) in Garut, West Java and Kuala Kurun-Central Kalimantan (representing a wet tropical climate) and Sumba-East Nusa Tenggara (representing a dry tropical climate); (4) the Yogyakarta Upland Agriculture Development Project (YUADP) at D.I. Yogyakarta in 1990–1998; and (5) the National Watershed Management and Conservation Project (NWMCP) in 1995–2000 [38].

## 4. Forty Years of SWC Implementation

In Indonesia, the development of SWC implementation has been quite dynamic. It has been impacted by a paradigm shift in response to changes in national policies and plans, which manifests in regulatory changes. The most significant issue is the shift from a top-down strategy in the implementation of SWC from the 1960s to the 1980s to a bottom-up approach. This is indicated by an increase in the active involvement of the community as a form of community participation.

### 4.1. Sloping Land Conservation

In general, SWC practices on sloping land can be grouped into two categories, namely civil engineering (mechanical) and vegetative measures. The mechanical measures must be carried out immediately to control erosion; however, vegetative measures must also be implemented as longer-term efforts.

### 4.1.1. Mechanical SWC Measures

Through the Presidential Decree on Regreening in 1976 (Inpres 8, 1976) and Inpres 1983, a national program was carried out to rehabilitate degraded land inside and outside forest areas using vegetative and mechanical methods. In 1984, a project, namely the Indonesia-Upland Agriculture and Conservation Project/UACP (Loan 2474-IND), aimed to increase farmers' production and incomes in Java's densely populated upland areas while minimizing soil erosion. The main component of this project was the construction and improvement of bench terraces and plant management practices [39,40]. Sinukaban [39] has reported the effectiveness of terraces, drop structures, drainage ditches, and riser's

covers on the steep slopes (8–35%) in controlling soil erosion. The UACP pilot scheme, which combined terrace improvement, cropping patterns and fertilization in Central Java, provided the benefit of increasing community incomes [41]. The project was ended in 1993, which resulted in the UACP-farming system. The project confirmed the difficulty of introducing technological change based on intensification because of the high-risk factor and labour constraints. In addition to terraces, some mechanical methods are *rorak* (silt pit), vertical mulch, rows of stones, drainage channels, and contour beds [42].

Various mechanical measures carried out in that period by the government through the Ministry of Agriculture and the Ministry of Public Works, as well as those carried out by foreign projects (Word Bank, JICA, etc.), were generally the construction of streambank retaining structures, gully plugs (Figure 4), retaining dams, and check dams for erosion control in ravines and reservoirs of water sources. Terraces, *gulud* (ridges), *rorak* (silt pit), grass barriers, and contour farming were applied on cultivated land. These techniques were usually combined with vegetative methods to prevent runoff while increasing infiltration.

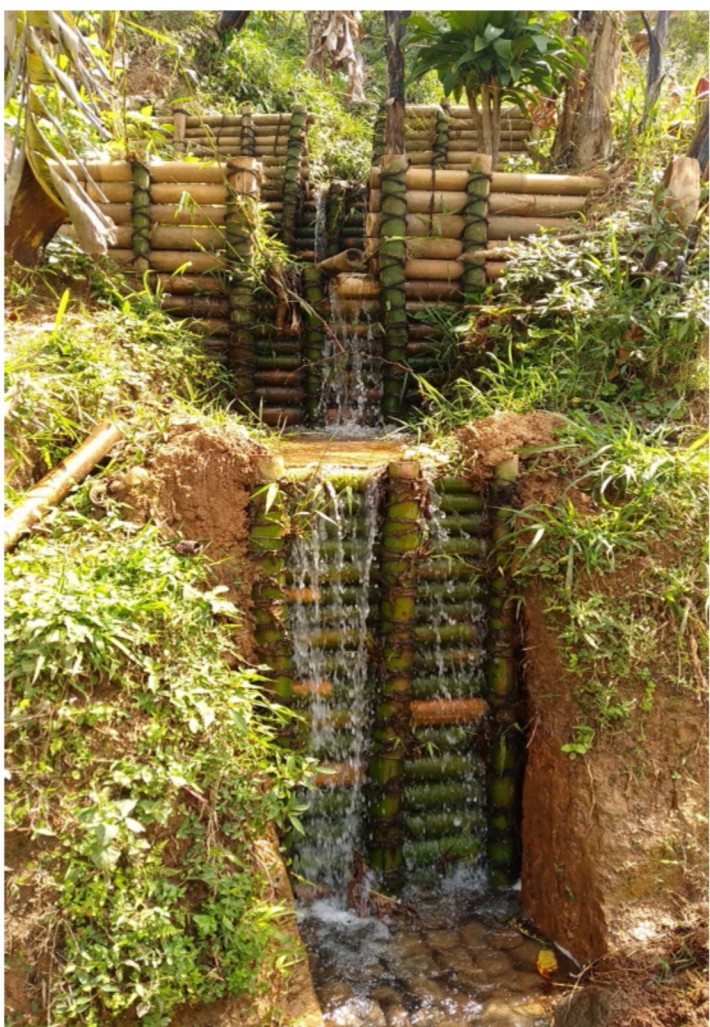

**Figure 4.** Gully plug construction from bamboo for controlling gully erosion.

Some of these activities were then also applied by the community, accompanied by the government and NGOs, using local resources that were simpler, easier to obtain, and cheaper, such as stone, wood chips, twigs, bamboo, and straw. The material was used for various terraces, channels, falls, and landslide retaining walls [43]. For controlling small gullies, bamboo was used as a base material, which is quite effective [43–46].

The implementation of making gully plugs and check-dams is regulated in the Regulation of the Directorate General of Watershed Management and Protected Forests No. P.6/PDASHL/SETKUM.1/8/2017 [47]. The government supports SWC activities to control trench erosion through the Ministry of Environment and Forestry. During 2015–2019, around 1.5 trillion rupiahs (3% of the total budget) have been allocated to construct gully plugs and check dams throughout Indonesia. These were carried out in the upstream area as part of degraded watershed recovery. Unfortunately, this government support has not shown success in reducing the rate of erosion and land degradation. Gully plugs cannot function as a ditch erosion control independently because the construction will experience a decline in function over a certain period. To effectively function, the gully plug needs to be supported by the massive application of SWC in the upstream area [48].

For the SWC implementation, the government needs to assist the community at the site level [9,49,50]. Community empowerment through micro watersheds is expected to ensure the sustainability of watershed management from upstream to downstream [20,49,51,52]. Participatory and collaborative planning for SWC techniques must be adapted to specific environmental, economic, and socio-cultural characteristics [27,49,51]. This plan increases community knowledge to utilize local materials for SWC construction, such as bamboo [20,53]. Since 1993, WMTC has developed bamboo as a small ditch plug to overcome ditch erosion and stabilize slopes in the Wonogiri Watershed, Bengawan Solo River Basin. This successfully reduced 52.26–57.95% of gully development. Another soil erosion control that empowers the community is a form of gully plug conservation that applies interlocked Lego bricks made from waste ash [54].

### 4.1.2. Vegetative SWC Measures

The promotion of vegetation as an approach to SWC methods is a convenient way to carry out the land rehabilitation process, both in the direction of reforestation (rehabilitation within the scope of state forest area) and regreening (rehabilitation within the scope of the private area) [32,55]. Furthermore, the implementation of vegetation techniques provides advantages in simplicity, efficiency, and low cost [56].

The SWC programs in Indonesia that focused on vegetation-based treatment were started in October 1951 when the Indonesian government launched a regreening program called the *Karang Kitri* (Figure 1). This program became the forerunner to Indonesia's campaign for land rehabilitation and community forest development. The target of this program was planting the Multi Purposes Trees Species (MPTS) in home gardens to increase land productivity and food security [32]. The Karang Kitri program was neglected due to the unconducive political situation and lack of financial support. Therefore in 1961, the regreening program was started again as the National Regreening Week (Pekan Penghijauan Nasional). This program expands the goals of the Karang Kitri not only to increase land productivity for increasing people's prosperity but also for preserving soil and water from floods, erosion, and landslides [57].

In 1967, the rehabilitation program began to be seriously carried out by combining planting and civil engineering methods through the Deptan program 001 and 002 under the Ministry of Agriculture, which targeted the Wonogiri Regency area and was expanded to the entire Surakarta Residency. It was a turning point in awareness of comprehensive rehabilitation efforts based on SWC after the flooding hit Solo. Furthermore, various rehabilitation programs were initiated continuously from local to national levels. The *Rakgantang* is an example of the regreening movement initiated locally at the provincial level in 1961 by the Governor of West Java. Meanwhile, the planning and implementation of the rehabilitation program are being improved and expanded nationally.

In 1983, rehabilitation began to develop, focusing on Java and others. Likewise, the rehabilitation targets are private land and state forest areas except for the Nature Reserves and core zone areas of the National Parks. The terms watershed and degraded soil have been used conceptually based on Instruction No. 5, 1976 and then developed contextually as a unit of analysing, planning, and implementation.

There are many types of vegetative rehabilitation methods in Indonesian watersheds as a form of SWC. Tree-based techniques are associated with reforestation and modifications, such as agroforestry or community forest, mixed gardens, silvopasture systems, alley cropping systems, and contour hedgerows [58]. Other non-tree-based techniques, such as grass barriers, mulching, cover crops, and living hedgerows systems, use low plants or crop residues to protect the soil from precipitation and runoff erosivity [20,58] (see Figure 5).

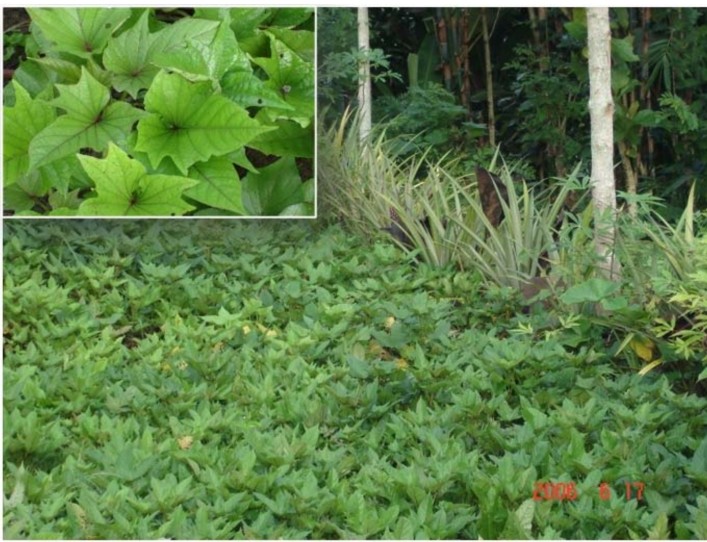

**Figure 5.** Sweet potato plants as ground cover for erosion control and forage for livestock combined with pineapple strips for terrace enforcement. Its fast-growing leaves are a favourite fodder for pigs and its bowl-like shape can absorb the kinetic energy of rain. Meanwhile, the stems and roots that stick out above the ground are able to hold the soil from erosion.

### 4.2. Soil and Water Conservation of Mangrove and Peatland

Wetlands International in 2009 found that Indonesia has approximately 40 Mha of wetlands [59]. The destruction of peatland has reached 3–4 Mha of drained peatland [60–62]. Presidential Decree No. 82, 1995, which stipulates the start of the Mega Rice Project (MRP) in Central Kalimantan to convert one million hectares of peatland and swamps into rice fields, was recognized as a mistake that ended in total failure and became a dark history of peat management in Indonesia [63]. Ten years later, in 2016, the government of Indonesia established the Peatland Restoration Agency, hereafter referred to as BRG RI, with the mandate to restore 2 Mha of Indonesian peatlands in seven provinces of priority.

The direct target for BRG RI restoration is the community's peatland areas (non-concession), conservation areas, and protected areas. In contrast, the license holders of concession areas should take responsibility under BRG RI and Ministry of Environment and Forestry (MoEF) supervision. The strategies of peatland restoration in Indonesia include rewetting, revegetation, and revitalization of local livelihood [63,64]. In 2020, BRG RI received the additional task of rehabilitating 600,000 ha of mangrove in nine provinces (and the institution name was changed to BRGM). This showed the Indonesian government's commitment to restore mangroves by setting an ambitious target to rehabilitate 600,000 ha by 2021–2024. Different from the peat restoration, mangrove rehabilitation uses three strategic approaches: restore–improve–maintain (*3M memulihkan–meningkatkan–mempertahankan*). Due to the COVID-19 pandemic, which started in 2020, more than 17,000 ha of mangroves have been replanted to give job opportunities for local communities [20].

According to ref. [65], Indonesia owns the largest mangrove areas in the world, approximately 3.2 million ha; however, most of these (approximately 70%) have been destroyed. Mangrove destruction increased in the 1970s in Java, Sumatera, and Kalimantan due to the increasing amount of timber production and aquaculture development [20]. Efforts to

rehabilitate and restore Indonesia's degraded mangrove areas have been carried out by replanting 30,000 ha through various national programs such as the Forest Rehabilitation Movement (GERHAN), One Man One Tree, One Billion Tree, and Kebun Bibit Rakyat (People Nursery) or KBR programs [65,66]. Companies such as Pertamina, Perhutani AEON, PGN, KNI, Bank Mandiri, and various NGO's such as Wetlands Indonesia, OISCA, and Yamamoto Foundation, and also international institutions and donors such as JICA, KOICA, ADB, ITTO, and many others have also contributed more than 50,000 ha and 900,000 seedlings to the mangrove rehabilitation program [65].

### 4.2.1. Rewetting

Rewetting is one of the methods for repairing the degraded peat since draining the wetland tends to damage peatland [67,68]. It was first implemented by Wetland International, WWF-Indonesia, and the CIMTROP around 2005 in the Mega Rice Project in Central Kalimantan [69].

The rewetting program of building canal blocks was carried out by BRGM in seven provinces, Riau, Jambi, South Sumatera, West Kalimantan, Central Kalimantan, South Kalimantan, and Papua [70]. In South Kalimantan, Fauzi, et al. [71] found that belangeran (*Shoreabelangiran*), gelam (*Melaleuca kajuputi*), pelawan (*Tristaniopsis* sp.), resak (*Vatica wallichii*), rengas (*Gluta renghas*), and bangkirai (*Shorea laevis*) are local trees that are suitable for constructing the dams. Canal blocking in South Sumatera and Jambi uses compacted peat to surround the canals [72]. The advantage of this method is that there is no need to use wood materials, and the holes where peat material is taken from can be used as fishponds.

### 4.2.2. Revegetation

The revegetation efforts on degraded peatland have been carried out concurrently with rewetting activities in Central Kalimantan since the 2000s [73]. In the Ex-Mega Rice Project (EMRP) of Central Kalimantan and Berbak National Park of Jambi Province, seedling nurseries have been established and a planting program implemented [74,75]. Moreover, after dam establishment at the primary canals of EMRP, Wetland International established seedling nurseries and planting to EMRP during the CCFPI program 2003–2005 [76]. Other seedling nurseries and planting activities have also been established during the Central Kalimantan Peatland Project 2008 at EMRP and Sebangau National Park of Central Kalimantan. However, it should be underlined that those revegetation activities implemented by various non-government organizations are mostly pilot and trial-based in terms of nature and scale [73,77,78].

The peatland restoration that has been conducted on a large landscape run by a private sector in production forests of Central Kalimantan, Indonesia, is the Katingan Peat Conservation and Restoration Project, or The Katingan-Mentaya Project (KMP). The KMP restoration program was established in 2013 and scheduled to run for 60 years (2013–2073) following the ecosystem restoration concession granted by the MoF, its main tasks include revegetation through planting at deforested areas, assisted natural regeneration, rewetting, and community-fire prevention. The lessons learned for revegetation at the KMP peatland area of Central Kalimantan concern the onset planting time, seedling selection, planting technique, protection, monitoring, and local community involvement, considering that the high cost of planting is predicted to be more than US$500/ha [73].

### 4.3. Water Resource Management

In Indonesia, an Integrated Water Resources Management (IWRM Principles) has been stated in Law No. 07, 2004. In fact, Indonesia has experienced four generations of laws related to water, which are the *Algemene* Water Regulation of 1936, Law No. 11, 1974 on Irrigation, Law No. 7, 2004 on Water Resources (although on 18 February 2015, its overall validity was cancelled by the Constitutional Court (MK)), and Law No. 17, 2019 on Water Resource replacing the Law No. 7, 2004.

In general, WRM practices are implemented in the frame of watershed management. WRM practices, in the frame of SWC implementation, differ among watersheds, upstream-downstream areas, urban-rural areas, and water-related issues. All the WRM practices must be tailored to the needs of, and harmonized with, national/regional/local spatial planning documents (RTRW, RPJMN, or RPJMND).

The water-related issues in WRM and SWC practices include flood, water shortage, and water quality [79,80]. Earlier flood control strategies in Indonesia mainly concentrated on developing and improving flood prevention infrastructures [81]. These efforts seem to be supported by efforts in reducing potential flood events through SWC implementation, especially in the upstream watershed. In Indonesia, SWC implementation has been carried out with benefits related to potential flood reduction through declining potential runoff, through methods such as the vegetative approach and mechanical SWC (terraces, infiltration ditch) [82,83]. An infiltration well has been implemented for urban and downstream areas, aiming to control flood potential by reducing surface runoff and recharging groundwater levels [84].

As an archipelago country, Indonesia has more than 10,000 small islands with an area of fewer than 10,000 m$^2$ as defined in the Minister of Marine Affairs and Fisheries Decree No. 41 of 2000 [85]. Watersheds of the small islands are characterized by a relatively short length of the main river, and the water mostly flows directly to the sea [86]. Water-related issues on small islands are water shortage and groundwater quality [87–91] due to limited water storage capacity [92], geological characteristics, lack of land cover [93], and narrow areas of rainwater catchment [86].

In addition to small islands, water shortages also occur in dry areas. Dry regions in Indonesia generally spread in the eastern part, such as Lesser Sunda Islands (Bali and Nusa Tenggara), most of South East Sulawesi, North Sulawesi, South Sulawesi, as well as in a few areas in Java, Sumatera, and Kalimantan [94]. These regions experience frequent drought; therefore, water is the limiting factor for various purposes [95]. The SWC practice that has been widely implemented is water harvesting from rainwater or surface runoff. In addition, indigenous knowledge related to water harvesting also exists as a reflection of adaptation patterns in water scarcity [96]. For rural areas at the upstream watershed, water harvesting techniques consist of the on-farm reservoir (known as '*embung*'), channel reservoir, check dams, and infiltration ditches [84,97]. These techniques provide several advantages, such as collecting rainfall and runoff, storing water in the rainy season for dry season use, collecting sediment, and supporting farming irrigation.

In recent decades, urban areas have also experienced water quality problems. About two-thirds of public water supplies are generated from contaminated surface waters [81]. According to a water quality assessment undertaken by the MoF in 2008, the majority of well-known rivers in Indonesia are polluted, such as the Musi River (Sumatera), the Mahakam River (Kalimantan), and the Citarum River (Java) [98,99]. Technically, upgrading the facilities to handle polluted water would be a viable alternative. However, it is a costly solution that only alleviates the problem until pollution levels exceed treatment capacity. Therefore, the long-term answer is to clean up pollution at its source [100]. In addition, the SWC practice considered as a potential wastewater treatment alternative is a constructed wetland or artificial swamp [101].

### 4.4. Community-Based SWC Approach

The paradigm shift from a top-down to a bottom-up approach is critical in understanding the dynamics of community participation in SWC in Indonesia. In the 1970s, the top-down approach was used in implementing the Presidential Decree on Regreening and Reforestation [102]. According to Dixon and Easter [103], the top-down approach generally fails because it ignores local community and farmer incentives. Putro et al. [104] stated that soil and water conservation efforts failed because the activities carried out only focused on physical conditions without paying attention to community conditions and without community participation. As a result, the technology applied is often inappropriate [6].

Considering the weaknesses of the top-down approach, in the 1980s and 1990s, efforts to rehabilitate land were in transition, which led to the concept of participation [105]. The concept of participation should involve the community from the planning and decision-making processes to evaluation. In practice, community participation is still passive participation. The community is only involved in implementation. Even if the community participates in the planning process, its role is only to provide basic information, while the government's analysis and decision-making processes are carried out [102,106,107]. This results in the failure of the sustainability of activities due to the absence of community independence.

Since the 2000s, community empowerment in SWC has received more attention. Some government programs related to SWC efforts carried out by various ministries such as (RHL), National Movement of Water Conservation Partnership (GN-KPA), Development of Integrated Land Conservation Farming (PUKLT), and Development of Micro Watershed Model (MDM), have posed community empowerment as the main activity [102]. The government's seriousness regarding community empowerment in SWC can also be seen from the various regulations issued. Law no. 37, 2014 on SWC and Government Regulation No. 37, 2012 on Watershed Management includes articles on community empowerment in watershed management. In addition, in 2014, the MoF also issued MoF Regulation No. P.17/Menhut-II/2014, which specifically regulates the procedures for community empowerment in watershed management.

In community empowerment, the community is the decision-maker, meaning that the approach used is purely bottom-up. The results of a study on community empowerment show that economic considerations very often influence the community in making decisions for land use so that the conservation aspect receives less attention [108,109]. Based on these conditions, community-based SWC approaches need compromise and integration during the decision-making process so that the SWC objectives and community needs can both be fulfilled [27,109]. In addition, the community can also explore their local wisdom because the Indonesian people have a lot of local wisdom related to SWC efforts which vary in each region [110,111]. The results of the study state that local wisdom also affects the success of community participation in soil and water conservation [112].

## 5. Research, Development, and Innovation 1980–2020
### 5.1. History of SWC R&D

Indonesia's history of soil erosion and soil conservation research began in 1905. It coincided with the establishment of the *Laboratory tot Vermeerdering de Kennis van den Bodem* (Laboratory for the expansion of soil knowledge), which is part of *Laboratoria voor Inlandschen en voor Bodemonderzoekingen* (Laboratory for smallholder agriculture and soil investigation) as a supporter of the Van Landbouw Department, which began to function in August 1905. However, more programmed and organized soil conservation research was developed in 1969/1970 to establish the Soil Conservation Section at the Soil Research Institute, Ministry of Agriculture. Chronologically, the historical outline of the development of soil conservation research can be divided into several periods as follows:

### 5.1.1. Period of 1970–1980

This period is marked by the start of cooperation between the Government of Indonesia and the World Food Organization (FAO, UN) during 1972–1978, in the form of "Project Management of the Upper Solo Watershed". This cooperation was undertaken in order to formulate a watershed management system through trial activities in various areas of the Solo Upper Watershed, activities which became recognized as milestones in the watershed management system in Indonesia. During this period, the development of science and technology and soil conservation research was dominated by activities in the laboratory and greenhouses, supported by several demonstration plot research activities in the field. Simulation and modelling techniques were also developed, such as a rainfall simulator, Universal Soil Loss Equation (USLE), and Revised USLE (RUSLE). From 1970 to 1979, research on SWC resulted in modified erosion estimation using the USLE model

for erodibility (K), crop factor (C), erosion rates on various agricultural lands, erosion control technology, and land rehabilitation technology [9]. Some of the main science and technology innovations produced during this period were (1) the value of the soil erodibility factor of Indonesian soil, (2) the value of cropping factors and erosion control measures, (3) the use of soil conditioners, (4) the level of soil erosion on various agricultural lands, (5) organic matter management technology, (6) soil management technology, (7) erosion control technology, and (8) soil rehabilitation technology [113].

5.1.2. Period of 1980–1991

In 1980, watershed management was introduced, including the integration of various sectors and local governments into SWC. In this regard, SWC is also a part of watershed management activities. The purpose of watershed management is the creation of continuous water yields both in quality and quantity, increased land productivity, and improved community welfare. Research on SWC since 1980 has increased rapidly; carried out both by the Soil Research Center and various foreign Cooperation Projects such as the Land and Water Forest Rescue Project (PHTA) carried out in some watersheds, including Citanduy, Jratun Seluna, Konto, and Bengawan Solo. At that time, several effective soil conservation techniques were produced from various field conditions. This period was also marked by the change of the Directorate General of Forestry into a Ministry of Forestry separated from the Ministry of Agriculture.

Given the importance of the tasks of the P3DAS Project, the project organization was changed to a structural institution through the Decree of the Minister of Forestry No. 098/Kpts-II/1984 under the name of Balai Teknologi Pengelolaan Daerah Aliran Sungai /Center for Watershed Management Technology (WMTC) and is the Technical Implementation Unit (UPT) of the Directorate General of Reforestation and Land Rehabilitation (Ditjen RRL), MoF.

The WMTC has produced various research findings. One of the phenomenal discoveries is the use of arc-shaped dams to control sedimentation. The arc dams were first tested by WMTC in 1986/1987 [114]. Until now, the arc dam is acknowledged as one of the SWC methods as Director General of Watershed Management and Protected Forest regulation number P.6/PDASHL/SET/KUM.1/8/2017 on Technical Guidelines for Soil and Water Conservation Buildings.

Various publications such as scientific articles, books, and technical guidelines have been produced and widely adopted in Indonesia. Some of these are the criteria for degraded land in Indonesia in 1982, which was revised in 1995 [115], watershed management with degraded land management in the Samin Watershed in 1988 [116], and trials of Gobeh Sub-Watershed management in 1989 [117]. In 1992, WMTC was assigned as a representative of Indonesia and, in collaboration with the New Zealand Land Resources Mapping Project, compiled a land resource survey handbook for soil conservation planning [118]. Referring to the handbook, assessments for soil conservation planning began to be carried out throughout Indonesia. In the next period, WMTC began to introduce the SWC method through various publications. Among these publications are arc-type dams construction (1996) [114], soil conservation in sugarcane cultivation in dryland (1996) [119], degraded land rehabilitation using bamboo (1996) [120], grass-barrier construction (1996) [121], and technical guidelines for measuring erosion in agricultural land (1997) [122].

5.1.3. Period of 1991–2020

In 1991, WMTC, which is mostly engaged in testing, reviewing, and developing watershed management technology, was transferred from the Technical Implementation Unit (UPT) of the Directorate General of Reforestation and Land Rehabilitation to become the UPT of the Forestry Research and Development Agency, in accordance with the Decree of the Minister of Forestry No. 171/Kpts-II/1991 dated 23 March 1991, with working areas throughout Indonesia. Based on the Decree of the Minister of Forestry No. 1048/Kpts-II/1992, the second WMTC was established in Ujung Pandang, South Sulawesi, with the

working area of Eastern Indonesia (KTI). With the establishment of the WMTC of Ujung Pandang, the first WMTC changed to WMTC of Solo with a working area of western Indonesia. During this period, a lot of research and development was carried out regarding SWC on specific lands such as agroforestry land, oil palm plantations, ex-mining land, and on SWC techniques at the micro watershed scale.

There are several projects in Indonesia which also include study/trial activities, including Research on Increasing Productivity and Soil Conservation to Overcome Shifting Cultivation, 1990–1993; Upland Farmers Development Project (UFDP) in West Java, Central Kalimantan, and East Nusa Tenggara, 1993–2000; Working Group on Research and Development of Dry Land Farming Systems in the Cimanuk Watershed, 1995–2000; Managing of Soil Erosion Consortium (MSEC) in Central Java, 1995–2004; and Agricultural Multifunction Research, among others to formulate agricultural development policies and land use, 2000–2005. The research and development activities resulted in various technologies and systems of conservation farming (SUT), including institutional models and dissemination systems [113].

During this period, WMTC also produced several guidelines related to watershed management, including SWC practice guidelines (2002) [123]; causes and solutions of floods (2002) [124]; pine forest and water products (2002) [125]; evaluation report of flash floods in Bukit Lawang, Bohorok district, Langkat Regency, and North Sumatra (2003) [126]; guidelines for monitoring and evaluation of watershed management (2004) [127]; technical guidelines for watershed management (2005) [128]; technical guidelines for creating erosion plots with collector tanks (2006) [129]; and technical guidelines for watershed monitoring equipment (2006) [130]. Since 2006, the direction of WMTC's R&D has focused more on watershed management to deal with land degradation and natural disasters. A quick assessment of sub-watershed degradation, published in 2006 and revised in 2010 and 2012, discusses land degradation and its relation to several hydrometeorological disasters [131]. This book was adopted in the Director-General of Land Rehabilitation and Social Forestry regulation No. P.04/V-Set/2009 on the monitoring and evaluation of watersheds. Furthermore, WMTC published a book on flood and landslide mitigation techniques (2008) [132], watershed management planning systems (2012) [131], and reservoir catchment area management systems (2014) [133].

Starting in 2015, WMTC began to develop catchment-based research and produced several research findings that were published nationally, including micro watershed management (2015) [134], Ciliwung watershed restoration (2015) [135], water-friendly trees (2016) [136], vegetative engineering for reducing landslides (2016) [137], gully erosion control (2018), integrated watershed management at the implementation level (2019) [138], and sustainable water resources management (2019) [139].

### 5.1.4. Period of 2020–Onwards

During this period, participatory action research was further enhanced in line with the paradigm shift in national development that placed the community as the subject of development. However, 2020 was also a year of transition heading to the operation of a new national R&D institution called *Badan Riset dan Inovasi Nasional*/BRIN (National Research and Innovation Agency), which integrates all research institutions into one institution. Great hopes are placed on the new institution to make SWC research more focused, needs-based, integrated, collaborative, and complete with a clear plan of results in the form of product prototypes, appropriate technology, and scientific publications of international standard.

### 5.2. R&D Policy Direction: Evolving Concept of Indonesian Soil Water Conservation Research & Development

The implementation of research on SWC is aligned with the national development policy in the field of science and technology. During the New Order era, the direction of science and technology policy was outlined in the Repelita. During the first and second

Repelita, government policy was directed at establishing and increasing the number of R&D institutions and improving research facilities and infrastructure. In the third and fourth Repelita, the policy was more directed towards the development of science and technology, prioritizing technology transfer, especially high technology, human resource development, and basic research implementation. SWC research activities in the Old Order were carried out through research policy programs such as National Strategic Excellence Research, Integrated Excellence Research, and Partnership Excellence Research [140].

In the post-1998 reform era, research and development policy was directed at internal strengthening, development, and diffusion of science and technology by protecting intellectual property rights and international cooperation. Research and development activities in the SWC sector under the coordination of the Ministry of Forestry have been directed by the National Long-Term Development Plan 2005–2025 and the strategic plan of the Ministry of Forestry. The SWC research was carried out to support the optimal management of 282 priority watersheds, including strategies for managing water catchment areas to protect vital objects [141]. The implementation was described based on the proposed research activity in 2005–2009 with the topic of technology and institutions for the rehabilitation of degraded lands. During this era, national-scale integrative research involving researchers at the central and regional levels was initiated.

Furthermore, the research and development plan under the Ministry of Forestry, including SWC topics, was arranged for five years. In 2010–2014, SWC's research policy direction referred to the forestry research and development roadmap (2010–2025). The research was integrated and carried out under the topic of land and water resource management supporting watershed management, while in the 2015–2019 period, it emphasizes water resource conservation [142].

In 2017, there was another change in national research and development activity policy direction in the issue of a national research master plan valid until 2045. The plan includes ten research fields, and their implementation is described in each five-year planning as a national research priority. In the 2020–2024 period, research related to SWC can be classified into multidisciplinary and multi-sectoral research focuses, with the themes of environmental research, water resources, and climate change [143].

With the enactment of the national research priority, the five-year research plan at the Ministry of Forestry for the 2020–2024 period is directed to support the forestry research road map and is adjusted to the national research priority focus. SWC research can be included in the integrative research group of climate change mitigation-adaptation, environmental quality management, and improvement of sustainable forest management. The dynamics of policy direction in the implementation of SWC research are likely to continue to develop following the integration of all research institutes into BRIN. It is hoped that this integration will advance SWC's research activities, although it will take time to adapt to the new institution.

With more integrated research institutions through BRIN, future research challenges related to SWC, apart from the classic problems of controlling land degradation, involve increasing land productivity and mitigating climate change by optimizing the use of advanced technology. The direction of research and development related to SWC in the future should be concentrated on:

a.  Developing techniques for measuring various parameters of SWC to automatically support soil preservation, land, and watershed functions (smart systems, such as the utilization of satellites, computers, etc.). From various water conservation techniques that have been applied, information is still needed on the effectiveness of each technique in increasing infiltration because the application of a certain technique specifically depends on climate, soil, and topography [144].

b.  Sharpening the parameters of SWC to create an appropriate monitoring system (high precision). Soil conservation strategies often adopt traditional practices, for example, the application of terracing, which has been known for thousands of years. However, soil conservation practices that are currently implemented still require effective

scientific research to find appropriate designs for its efficient achievement [145]. For example, applying a terracing system must consider the proper dimensions, characteristics of the soil and topography, climatic conditions, the availability of supporting materials, and the objectives to be achieved [144].

c.　Make more detailed research related to the socio-economic characteristics of the community to formulate an approach so that the community understands the function of soil and land that must be managed sustainably. The application of soil conservation mostly only pays attention to the biophysical aspect. The lack of socio-economic and community culture consideration often results in low acceptance of farmers or land users [146]. Studies need to be conducted to explore the socio-economic and cultural aspects of the community that can encourage acceptance of soil conservation measures and understanding as to what obstacles may be encountered in its implementation [2,147].

d.　In implementing SWC technologies, attention to the institutional approach should be improved. Therefore, research related to institutional and community support related to land management, especially on degraded watersheds, should be undertaken. To effectively implement an SWC, political support is needed in the form of laws and regulation products agreed upon by all stakeholders, including the land user community [2]. Currently, research is still needed relating to increasing active community participation in the implementation of SWC [6]. The involvement is not only for government stakeholders but also the private sector and continues to encourage the implementation of payment environmental services arrangements. Appropriate policy support through legislation needs to be carried out through a study so that the obstacles in government authority distribution in the implementation of the SWC can be bridged and supported by government officials at the site level.

### 5.3. State of the Art of SWC R&D in Indonesia

### 5.3.1. R&D of SWC at Dryland Areas

R&D activities of dryland between 1980 and 1990 consisted of agricultural research programs to support transmigration program (P3MT), Crop Animal System Research Project, Upland Farmers Development Program (UFDP), Citanduy II Program, Dry Land Agriculture Program, Soil Conservation (P2LK2T), and the National Watershed Management and Conservation Project (NWMCP). In the period of 2000–present, R&D included Recapitalization of The Soil Fertility of Acid Upland Soils in Indonesia with Phosphate Rock, Program for Improving the Income of Poor Farmers Through Innovation (P4MI), PRIMA TANI, Consortium for Integrated Agricultural Systems in Dry Climate, Carbon Efficient Agriculture Systems, Multifunctionality of Agriculture and Soil Erosion Management Consortium [148].

Land degradation due to rainwater erosion is one of the main problems in the dryland. R&D related to SWC has been conducted and developed by various institutions to increase dryland productivity through controlling erosion processes. It can be categorized into two groups based on altitude (upland and lowland) and four groups based on the type of activity (improvement of soil properties, erosion control, land rehabilitation, and water conservation and management).

R&D of SWC in upland areas is commonly carried out to control erosion in the form of vegetative, mechanical conservation, and a combination of both [149,150]. Other SWC techniques offered for upland areas include conservation farming models and agroecology-based agricultural development [151]. Meanwhile, in the lowland, the R&D of SWC varies, including mechanical conservation techniques (terraces and silt pits) and vegetative techniques (alley cropping/alley cultivation, strip cropping, agroforestry, and ground cover crops). In several locations, SWC technology has been developed based on indigenous knowledge, such as "sengkedan" (Java and Bali), "tabatan watu" and "kebekolo" (East Nusa Tenggara), and live fences (West Nusa Tenggara) [148].

Improving soil properties research using soil conditioner has been carried out in the dryland. This research aims to repair damaged soil, and maintain and increase soil productivity for sustainable utilization [152]. Natural materials are mainly used as raw materials for soil conditioners, such as organic materials and manure. Several studies in dryland regarding soil conditioners include the application of hydrogel [153], manure [154,155], biological fertilizer (mycofer) [156], and biochar [157,158]. Improvement of soil properties is also achieved by developing appropriate tillage techniques (mechanical) or vegetative as well as chemical measures. A vegetative method that has been widely studied in the improvement of soil properties is the implementation of mulch [159–162]. Many chemical methods for soil and water conservation have been carried out using synthetic or natural chemical preparations (soil improvement agents) [159].

Erosion control research activities comprise erosion prediction models (methods) and the study of erosion control techniques. In order to develop the USLE model, several studies obtained the values of the USLE factors, such as the index of R (erosivity) [163] and K (soil erodibility) [164]. Process-based hydrological models have also been applied, such as the RUSLE-GIS [165], AGNPS [166–169], ANSWER [170], GenRIVER [171], and SWAT model [172,173]. Research on erosion control techniques is mainly carried out to improve existing technology, including terracing [155], hedgerow crop [174,175], mulching [176,177], silt pit [178,179], grass strip [180,181], and vegetative technique [182]. The results of these studies are often used to inventory the level of erosion hazard, land use planning, and the selection of alternative SWC techniques. The SWC techniques have also been implemented in land rehabilitation research in the dryland. For example, the application of vegetative SWC on mined post-pumice [182] and degraded land of the dryland [183,184]. In addition, research on soil conservation and water management has been performed by developing water harvesting techniques, including rainwater reservoirs [176,185–188], and various irrigation techniques, such as drip irrigation [189,190].

However, various SWC at the farmer level face numerous obstacles due to several limitations, including technological and financial limitations. Therefore, it is a challenge for future SWC work to establish more adoptable techniques using local resources and indigenous knowledge. In addition, excellent and targeted planning is required for conservation techniques to be applied appropriately, effectively, and efficiently based on biophysical, socio-economic, and cultural factors.

### 5.3.2. R&D of SWC in Peatland Area

The establishment of canal blockings is one of the strategies in water management improvement of the peatland ecosystem [64]. The canal blockings are predicted to raise the water level, reduce the water flow out, and maintain the water level resulting in the improvement of peatland hydrological conditions. The selection of canal blocking types depends on the biophysical condition, canal dimension, peatland topography, the availability of materials, and accessibility to the location of planned canal blockings [72]. Canal blockings could effectively rewet the peatland up to 100 m on either side of them [72,191].

Revegetation can be carried out in two ways: passive and active revegetation [63]. Passive revegetation relies on natural regeneration. In order for this to work, seed rain and seed dispersal have to be sufficient for natural regeneration. Seed rain is used to enhance the supply of endemic plant natural seeds from degraded and adjacent pristine forests [192,193], while dispersers, such as natural and artificial bird perches, play important roles in promoting natural regeneration and regrowth due to their ability to enhance seed rain and distribute seeds [75].

Three main aspects determine the success of active revegetation or assisted regeneration [63]. Those are the availability and high quality of seedlings, appropriate planting techniques, and post-maintenance after planting. A list of 26 local peatland species and their ecological tolerance to extreme environmental conditions, conditions required for optimal growth, and implications for restoration has been released [194]. From the list, we can opt for species that adapt well in shade conditions, species that adapt well in inundation

conditions, or species that adapt well in an open area. Planting techniques by peat chopping before planting and compacting soil surrounding seedlings, establishing mounding, and using seedlings with the height of 50 cm or higher can increase the survival rates of planted seedlings in inundation conditions [195]. One innovation to overcome low access of plant roots to nutrients and oxygen in the waterlogged condition is the Aero Hydro Culture technology [196]. This technology promotes aerial roots in the first 6–12 months after planting. The initial evaluation of the *Shorea belangeran* plantation showed that this technology has potential. Mycorrhizal fungi application is also important for root stimulation in inundation conditions [197,198].

In order to evaluate the success of degraded peatland restoration, certain criteria and indicators have been formulated [63]. Ecosystem function, peat-soil improvement condition, hydrological condition, the diversity of flora and fauna, carbon stock, and community livelihood improvement are the criteria that shall be looked at when we want to evaluate the success of degraded peatland restoration.

### 5.3.3. Agroforestry as an Integrated Soil and Water Conservation Measure

Agroforestry was first recognized and researched on a national scale in the 1980s, focusing on local level agriculture, existing agroforestry composition characteristics, and interactions between tree-soil-agricultural crops in an alley cropping system that can increase soil fertility, control erosion, and increase crop production [199,200]. In the following decade, agroforestry research focused on understanding processes by testing hypotheses and predictions, strengthening theories with simulation tools, and shifting biophysical research from the plot to the landscape scale [200]. Research on agroforestry systems continues to develop along with environmental problems such as changes in land use, water scarcity, food security, bioenergy, global warming, climate change, and biodiversity [200,201].

As an integral part of a multifunctional working landscape, agroforestry delivers a number of ecosystem services and environmental advantages (carbon sequestration, biodiversity, soil enrichment, water, and air quality) in the soil and water conservation context [202]. Agroforestry can improve soil quality and health by increasing soil organic content (which is better than monoculture crops), increasing soil nutrients, soil fertility, and soil microbial dynamics (which is influenced by the presence of trees in the system) [203].

The presence of trees in agroforestry systems contributes to various agro-ecosystem services, including water retention, improved soil structure and porosity, increased organic content as a source of food for organisms, an increased presence of beneficial soil biota, and improved microclimate [204,205]. A combination of deep-rooted trees with fine root grasses can strengthen hillsides and stabilize river banks, thereby reducing the risk of landslides [206]. Therefore, tree selection, by choosing species that can grow adaptably in the cropping complex, is one of the keys to success in agroforestry [201,207]. In addition, climate change, floods and droughts, market dynamics, and poverty all bring opportunities and challenges for agroforestry research to generate knowledge and technology with strategies that lead to ecologically friendly systems and higher land production.

## 6. Bridging the Gap

Soil conservation policies in Indonesia have been in place since the 1970s, and research activities related to SWC started in the early twentieth century. However, Indonesia's SWC practices have not been fully effective. In addition to limited resources, this problem is partly due to the neglect of scientific evidence to formulate SWC policies.

## 6.1. Research–Policy Gap

In Indonesia, the use of evidence as a new paradigm in current policy-making is considered very important and opens up great opportunities for researchers to be actively involved in policymaking through collaboration with policymakers. However, efforts are needed to ensure that research findings are accessible and can be translated into practical policies by policymakers. Not all research is appropriate as a source of evidence for the policy-making process. There are several challenges faced in producing valuable research for the development of evidence-based policy (EBP) in Indonesia [208], among these are:

a.      Limited funding sources

In Indonesia, of the total state expenditure in the 2021 state budget (APBN), which reaches IDR 2750 trillion, the allocation for research is only IDR 9.9 trillion or 0.36%, still far from the minimum ratio of 1%. Based on data from the Ministry of Research and Technology, in 2019, Indonesia's research funding for GDP was only 0.2%, slightly higher than Vietnam, which is 0.19% of their GDP, but lower than Thailand, Malaysia, Singapura, Korea, and Japan which are, respectively, 0.39%, 1.1%, 2%, 4.2%, and 3.5% of their GDPs [209]. Under these conditions, the research challenge is to produce quality research with effective and efficient use of budgets.

b.      Supporting data constraint

Law No. 14, 2008 on Public Information Disclosure has been enacted, but in reality, it is not easy to access data from various agencies. In addition to accessibility issues, the quantity, quality, and continuity of the data (time series) are often not in accordance with the needs. The government has begun to resolve issues related to data, including the issuance of Presidential Regulation (Perpres) No. 9, 2016, on the Acceleration of the Implementation of the One Map Policy.

c.      Non-linearity of research and policy needs

The current problem is that the research topics produced are not in-line with the needs of policymakers due to the lack of communication and coordination between research providers (researcher, scientist) and research users (policymakers). Research results do not match the needs of policymakers.

d.      The gap between research language and policy language

The research, in general, is too scientific and cannot be easily and precisely stated by policymakers in policy norms. The complexity of research often does not match the way policymakers think. In fact, transforming the resulting knowledge into effective policy solutions remains a question for many parties [210]. Choi, et al. [211] have stated that some strategies could be used to bridge the gap between science and policy, including science-policy forums, policy briefs, policy recommendations, joint research projects, conferences, personal contact, and collaboration in analysis.

## 6.2. Policy Implementation Gap

Problems in implementing laws and regulations are not only related to limited money and resources. Instead, these involve a complex set of factors that need to be addressed, such as orientation, incentives, and capabilities/capacities of the implementers [212].

At the operational level, most technical regulations in Indonesia are normative. In general, regulations are developed based on standards and guidelines or opinions of policy-makers and do not consider scientific principles that are actual, objective, and testable. The problems are thus how to implement the regulation and how to translate the regulation into a lower order and more practical regulation, such as technical guidance of site management. Figure 6 shows an example of an evidence-based policy formulation process at the technical level, adapted from Nugroho, et al. [213].

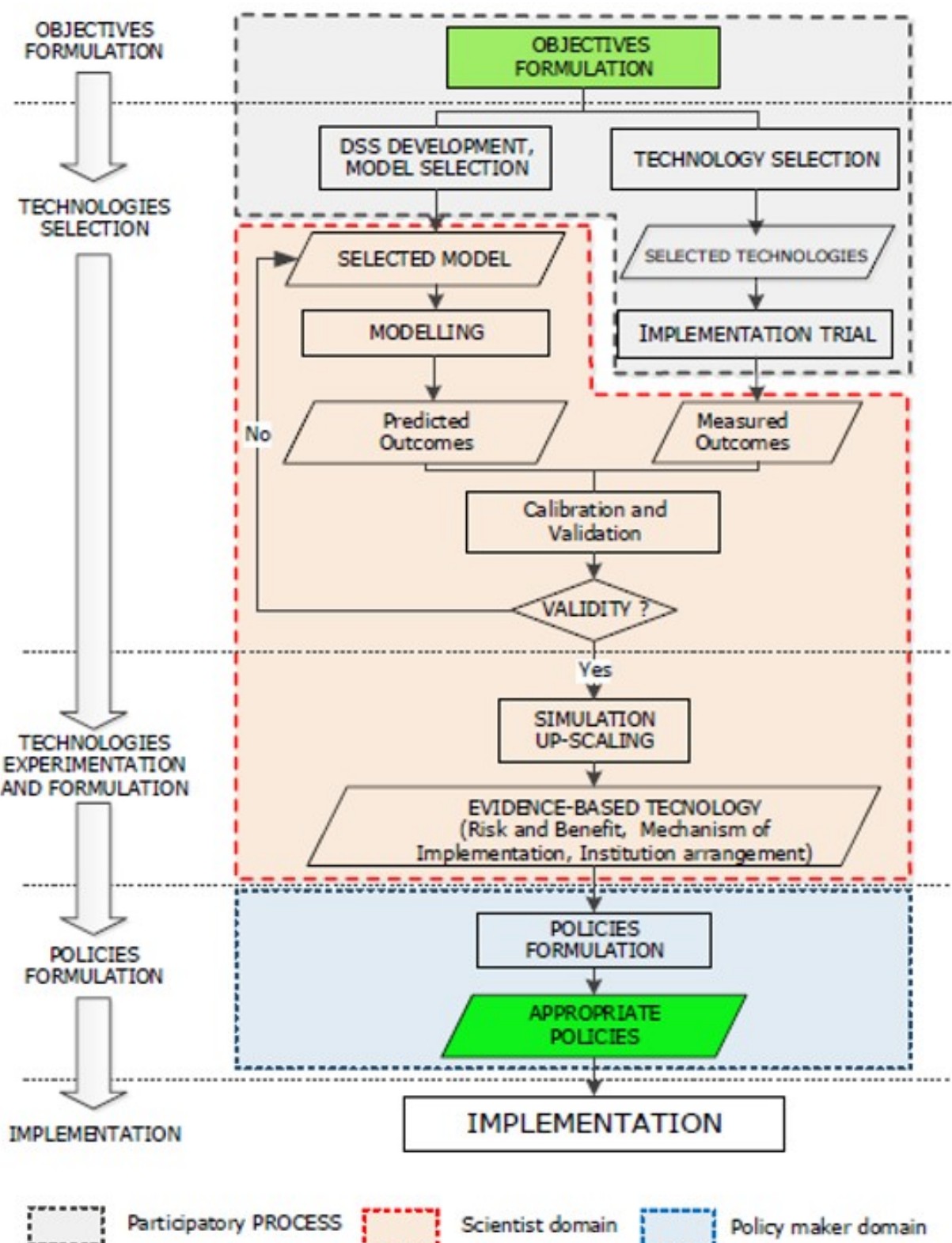

**Figure 6.** Framework for the formulation of evidence-based policies (adapted from Nugroho, van der Veen, Skidmore and Hussin [213].

An important factor for good connectivity between research, policy, and implementation is the balance between the complexity of real-world problems and the importance of meeting scientific standards, which in implementation can be translated into easy, simple, and straightforward language [210]. Regarding the flow of data and information needed in

research planning and policy formulation, it is necessary to formulate data and information traffic that flows smoothly between stakeholders. Based on the authors' experience as researchers and their observation of policy formulation dynamics in Indonesia, we formulate a scenario of data and information flow among stakeholders (policymakers, civil society, scientists and academics, legislators, and technical implementing units and practitioners) to realize EPB as depicted in Figure 7.

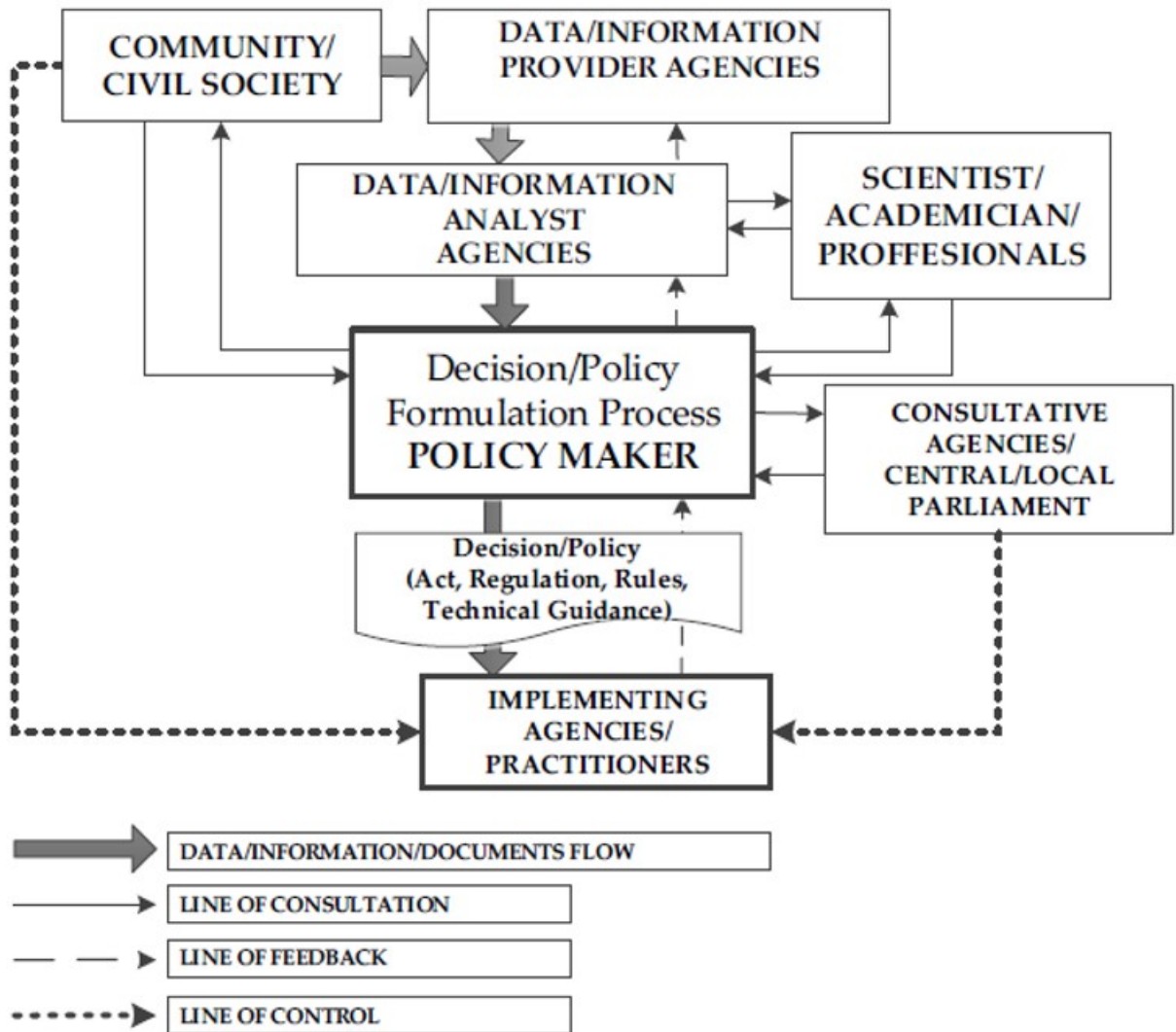

**Figure 7.** Concept of data and information flow between parties to optimize EBP.

Law No. 11, 2019 on the National System of Science and Technology, enacted on 13 August 2019, is considered a milestone for Indonesia's evidence-based policies development. This law contains a national system that regulates the pattern of relations between institutional elements and resources to build a network of science and technology as a unified whole in supporting the implementation of science and technology as a scientific basis in the formulation and determination of national development policies. One of the important articles in this law related to EBP is the article that stipulates that the results of research, development, assessment, and application must be used as a scientific basis in the formulation and determination of national development policies. EBP is obligatory, not voluntary.

One of the important follow-ups of Law No. 11, 2019, is the establishment of BRIN. The establishment of BRIN is one of the milestones in the Government of Indonesia's policy to support the implementation of science and technology as a scientific basis in the

formulation and determination of development policies. This means that evidence-based policy must be applied in the formulation and determination of policies in Indonesia [214]. With the establishment of BRIN, research budgets that were previously spread across various ministries/agencies will be integrated. Research integration will cover the entire management process, budget, and human resources [215]. Regarding the funding, the government has also issued a policy on allocating endowment funds for research, development, study, and application activities. Therefore, R&D activities can produce discoveries and innovations to support sustainable national development, improve the quality of life and welfare of the community, and can also increase self-reliance, competitiveness, and national attractiveness in advancing the nation's civilization through international relations [214].

The difference between BRIN and the Research Institute under the existing Ministry is the existence of a steering board in its organizational structure. At this time, at the beginning of its existence, the BRIN Steering Committee consists of 10 people, chaired by the Chairman of the Steering Committee of the Pancasila Ideology Development Agency (BPIP) and two deputy chairpersons, who are the Minister of Finance and the Minister of National Development Planning (PPN)/Head of National Development Planning Agency (BAPPENAS). With such a structure, coupled with the fact that the current chairman of the BRIN Steering Committee is the chairman of one of the largest political parties in Indonesia, there is great optimism that the gap between research, policy, and implementation, which has been a chronic problem for national development, will be eliminated. However, some parties also argue that the presence of political figures on the BRIN steering board will make this research institution not independent and become a tool for the political interests of certain parties.

## 7. Closing Note

The question that often arises from the general public, academics and legislators concerning decades of SWC activities is how successful the activities are regarding the indicators of quantitative improvement of watershed conditions and reduction of degraded land area over time. In fact, the existing data series cannot be compared correctly and reliably due to several changes in the way degraded land is defined. Since the 80s, there have been five changes to the regulations regarding degraded land criteria. The last of these is The Regulation of Director General of Watershed Management and Protection Forest Management No. P.3/PDASHL/SET/KUM.1/7/2018 on the Preparation of Spatial Data on Degraded Land, which is a derivative of Law 37, 2014 on Soil and Water Conservation. Using this criterion, which drastically changes the previous criteria, KLHK revised the area of degraded land from 24.3 million hectares to 14 million hectares (Figure 2). As with the degraded land criteria, criteria for determining watersheds that need to be addressed immediately have changed several times. The Joint Decree of the Three Ministers, namely the Minister of Home Affairs, the Minister of Forestry, and the Minister of Public Works Number: 19, 1984, No: 059/Kpts-II/1984, and No: 124/Kpts/1984 dated April 4, 1984, established 22 "super-priority" watersheds throughout Indonesia. In 1999, the Decree of the Minister of Forestry and Plantations No. 284/Kpts-II/1999 dated May 7, 1999, recorded the number of priority watersheds as much as 472 watersheds but no longer used "super-priority" instead adopting an order of priority, namely "priority I," "priority II" and "priority III." Most recently, in PP 37, 2012 on Watershed Management, the classification of watersheds no longer uses the term "priority watersheds," but "watersheds that need to be restored and watersheds that need to be maintained their carrying capacity." This makes it difficult for the public to test the success of watershed management and land rehabilitation, and soil and water conservation (LR&SWC) based solely on forestry statistics books.

Regarding the achievement of LR&SWC activities, From Figure 3 in chapter 2, what is interesting is that the reduction in the degraded land area from 2004 to 2018 is much larger than the accumulated area of land rehabilitation activities carried out by the Ministry of Environment and Forestry. Assuming the criteria used to map degraded land are the same as the criteria used to determine the location of RHL activities, it is suspected that

there are LR&SWC activities outside the formal activities of the MoEF that are successful and significantly reduce the area of degraded land. One of the programs that we believe has a significant impact on increasing the success of LR&SWC encompasses the Natural Resources Conservation Business (UPSA) and Permanent Agricultural Business (UPM) pilot units which were launched by the government in the early decades of the Ministry of Forestry establishment. Implemented from 1990 to 2001, UPSA-UPM were developed as pilot units to increase public awareness and participation in LR&SWC efforts. Referring to the Circular Letter of Director-General of PDASHL No. 6, 2019, dated 27 September 2019, regarding the implementation of RHL activities through the UPSA model, there are so-called blocks and affected areas. The block is UPSA–UPM demonstration plot (demplot); 10 ha for UPSA and 20 ha for UPM. The affected area, the area outside the pilot unit that is targeted to be affected by the activity, has an area of approximately 100 ha with a block area within it. This program was developed in 25 provinces through the Presidential Instruction for Special Assistance Program (Bansus). After that, the UPSA-UPM program was financed through the Special Allocation Fund (Dana Alokasi Khusus/DAK) and implemented by the district government. From 1990/91 to 2000, 9705 units were successfully developed [216]. In other words, 1.1 million ha of land were successfully rehabilitated during this period. The accumulation of the affected area, which is actually a replication of the demonstration plots up to 2001, is thought to have caused the area of degraded land to decline from early 2004 onwards. From the results of research conducted by Hudaya et al. [217], Katharina [218], and Puslitbangtanak [219] in several locations in Java and Sumatra, this assumption has been proven to be true. The adoption of LR&SWC activities from the UPSA-UPM model by the community is very high. Even some community forest areas in Central and East Java, such as in Gunung Kidul, Sukoharjo, Wonogiri, Ngawi, Ponorogo, Blitar Regencies have received ecolabel certificates [29]. However, to prove it on a wider scale, there needs to be more in-depth studies through a combination of remote sensing and ground checks. In addition to UPSA-UPM, several other SWC activities that contribute to the improvement of degraded land are carried out by the Ministry of Agriculture, NGOs, and various community groups using a self-help format, drawing on government funds, and/or utilising other funding sources such as CSR and foreign grants.

From the above, there are several important notes. LR&SWC activities must start from the "upstream." Upstream is not only in the context of space but also in the context of management. From this context, the meaning of "upstream" starts from the availability of valid data and evidence as a basis for planning. Data integration is needed to monitor the success of LR&SWC at district and national levels. The development of the LR&SWC method should be adapted to the unique site conditions for each location. Further, LR&SWC location's target and watershed prioritization should be based on standardized and consistent criteria. The next "upstream" concerns the perpetrator. Learning from the success of UP UPSA-UPM, involving as many parties as possible will increase success and accelerate the achievement of LR&SWC goals. Gully plugs, retaining dams, control dams, and reservoirs built upstream will quickly lose their function if the farmer's land is not cultivated with good LR&SWC principles. In this context, subsidies policy to farming communities to carry out SWC activities on their land are important. Finally, included in the "upstream" context is to provide the understanding and to strengthen literacy about watersheds, LR&SWC, forest benefits, risks of flooding, landslides, and droughts, and others, as early as possible to the community by including this activity in the local content curriculum at the elementary school level located in upstream watersheds to urban areas, especially those living in areas prone to erosion, flooding, and landslides. The development of a culture of environmental awareness and disaster alertness must start from an early age.

The notes above are important due to the challenges ahead. Indonesia has ratified the Paris Agreement on global efforts to reduce carbon emissions. In the updated nationally determined contribution (NDC), a national commitment to a global agreement to reduce GHG emissions, integrated with watershed management, is one of the main programs to be achieved. This would be achieved through two strategies, namely: (1) enhancing synergy

across sectors and regions in watershed management, and (2) mainstreaming/integrating climate change adaptation in watershed management to reduce risks/loss as a result of climate-related natural disasters [220]. In addition, SWC activities that have been carried out in Indonesia for a period of 40 years are expected to be a bridge for the achievement of SDGs, especially goal number 6 (clean water and sanitation), 13 (climate action), and 15 (life on land). SWC, especially the vegetative method, is in line with the objective of the United Nations Convention to Combat Desertification (UNCCD). As one of the participating countries, the activities of SWC strongly support this goal. This is not easy because, at the same time, in accordance with the mandate of law No. 11, 2020 on Job Creation, Indonesia encourages investment to improve community welfare, including through simplification of permits for the use of forest areas for economic activities.

## 8. Conclusions

The implementation of SWC in Indonesia for forty years has not experienced significant changes in terms of methods and techniques; however, there has been a shift in goal orientation, which is not only for controlling erosion and land degradation but also for improving the community's economy. In applying SWC, various challenges are faced in regulations, institutional aspects, policies, and implementations of SWC as well as research activities. The highest regulation in SWC, Law No. 37, 2014, has not been followed up in government regulations. In addition, the SWC activities are part of watershed management. However, the SWC is regulated by law. In contrast, watershed management is regulated at the government regulation level. Among the existing institution, there is a lack of synergy and more pronounced egotism separating the different sectors in the planning and implementation of SWC. Although implementation of the bottom-up approach has been attempted, the concept of participation has not been fully implemented in several government-based project.

In response to the above challenges, some efforts should be conducted. Regarding Law No. 37, 2014, more detailed regulations are needed to be implemented through government and ministerial regulations. This will make it easier for field implementers to carry out their duties in SWC. For the institutional problems, it is necessary to include SWC activities in local government development plans so that the relevant sectors will use SWC activities as references in preparing their work plans. To achieve sustainability in SWC implementation, the government needs to assist communities at the site level, and it is necessary to apply a cross-subsidies mechanism between upstream and downstream.

The SWC challenges are also observed in the research activities. The current problem is that the research topics produced are not in-line with the needs of policymakers. In addition, gaps in translating scientific research findings into applied technologies are barriers to communication between researchers and the community as well as policymakers. Despite facing many challenges, efforts to bridge research and policy continue. Law No. 11, 2019 on the National System of Science and Technology, enacted on 13 August 2019, stipulates that the results of research, development, assessment, and application must be used as a scientific basis to formulate and determine national development policies. The EBP is obligatory, not voluntary. As a follow-up to the law, the establishment of BRIN is one of the milestones of the Indonesian government's policy to support the implementation of science and technology as a scientific basis in the formulation and determination of development policies. There is great optimism that the gap between research, policy, and implementation, which has been a chronic problem for national development, will be eliminated.

**Author Contributions:** H.Y.S.H.N., T.M.B., I.B.P., E.S., P. (Purwanto), D.R.I., N.W., R.N.A., Y.I., A.B.S., P.B.P., D.A., E.P., T.W.Y., P. (Pratiwi), B.H.N., A.S., W.H., O.S. and R.N. had an equal role as main contributors who equally discussed the conceptual ideas and the outline, provided critical feedback on each section, and helped shape and write the manuscript. All authors have read and agreed to the published version of the manuscript.

**Funding:** This research received no external funding.

**Institutional Review Board Statement:** Not applicable.

**Informed Consent Statement:** Not applicable.

**Data Availability Statement:** Not applicable.

**Acknowledgments:** We thank anonymous reviewers for their detailed comments and corrections.

**Conflicts of Interest:** The authors declare no conflict of interest.

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
