# Peer review of "Forty Years of Soil and Water Conservation Policy, Implementation, Research and Development in Indonesia: A Review"

_sustainability, doi:10.3390/su14052972_

Round 1

Reviewer 1 Report

Please see attached comments

Author Response

Dear Reviewer# 1
Thank you very much for the comments and suggestions.
Attached is our response to your comments and suggestions.

Reviewer 2 Report

It seems to me an interesting descriptive work of political aspects, research and development, as well as the implementation of efforts to manage priority hydrographic basins by the Government of Indonesia.
Perhaps it is a bit extensive in its writing and includes excessive bibliography.
In general, I think that the wording should be more specific, offer new solutions and avoid repetitions, such as the first paragraph of the conclusions

Author Response

Dear Reviewer# 2
Thank you very much for the comments and suggestions.
Attached is our response to your comments and suggestions.

Reviewer 3 Report

The paper is written as a review of soil and water conservation over the last 40 years in Indonesia.

I find the article interesting, it contains everything that should be in the review and I found answers to everything I needed to know. At my discretion, I recommend adding a photo of some SWC measures that are mentioned in the text.

Author Response

Dear Reviewer# 3
Thank you very much for the comments and suggestions.
Attached is our response to your comments and suggestions.

Round 2

Reviewer 1 Report

The authors have responded to the reviewer suggestion adequately. The content of the manuscript has been revised